# Characterization Studies on the *sugC* Gene of *Streptococcus suis* Serotype 2 in Adhesion, Invasion, and Virulence in Mice

**DOI:** 10.3390/vetsci11090447

**Published:** 2024-09-21

**Authors:** Zhimin Dong, Cheng Li, Xiangxue Tian, Xiaoran Guo, Xiuli Li, Weike Ren, Jingjing Chi, Li Zhang, Fuqiang Li, Yao Zhu, Wanjiang Zhang, Minghua Yan

**Affiliations:** 1Tianjin Key Laboratory of Animal Molecular Breeding and Biotechnology, Tianjin Engineering Research Center of Animal Healthy Farming, Institute of Animal Science and Veterinary, Tianjin Academy of Agricultural Sciences, Tianjin 300381, China; zhimin13@126.com (Z.D.); keleaidisheng@163.com (C.L.); tiantjxms@163.com (X.T.); guoxiaoran1995@126.com (X.G.); renweike2000@126.com (W.R.); zhaojing_7590@163.com (J.C.); rookiezhang1979@126.com (L.Z.); lifuqiang2208@163.com (F.L.); 2National Data Center of Animal Health, Tianjin 300381, China; 3Institute of Agro-Product Safety and Nutrition, Tianjin Academy of Agricultural Sciences, Tianjin 300381, China; lxl5084@sina.com; 4State Key Laboratory for Animal Disease Control and Prevention, Harbin Veterinary Research Institute, Chinese Academy of Agricultural Sciences, Harbin 150069, China; yaozhu922@163.com (Y.Z.); wanjiang09@126.com (W.Z.)

**Keywords:** *Streptococcus suis* serotype 2, *sugC gene*, adhesion, invasion, pathogenicity

## Abstract

**Simple Summary:**

As a zoonotic pathogen, the exploration of new virulence-related genes and the functional identification of *Streptococcus suis* are of positive significance in revealing the pathogenic mechanism of the strain and developing new vaccines. This study provides evidence that the *sugC* gene is a virulence-related gene of the *Streptococcus suis* serotype 2 strain and plays a crucial role in regulating its adhesion and invasion. The *sugC* gene is a coding gene for the ATP-binding transporter-associated protein, which has been reported to exist in the highly virulent *Streptococcus suis*. In order to reveal the effect of the *sugC* gene on the virulence of *Streptococcus suis* serotype 2, TJS75, a wild-type strain, was used as a parent strain, and a knockout *sugC* strain and complementary strain were successfully constructed via homologous recombination technology. We found that the adhesion and invasion abilities of the knockout *sugC* strain in PK-15 cells were decreased. Most importantly, the knockout of the *sugC* gene could reduce the incidence rate, mortality and LD_50_ value of TJS75 in BALB/c mice. These results provide a reference for us to further study the pathogenic mechanism of *Streptococcus suis.*

**Abstract:**

The *sugC* gene of *Streptococcus suis* (*S. suis*) is a coding gene for the ATP-binding transporter-associated protein with strong pathogenicity. In order to reveal the effect of the *sugC* gene on the virulence of *S. suis* serotype 2, a wild-type strain of TJS75, isolated from fattening pigs’ brain tissue samples, was used as a parent strain, and a knockout *sugC* gene (Δ*sugC*) and complementary strain (CΔ*sugC*) were successfully constructed via homologous recombination technology. The biological characteristics of TJS75, Δ*sugC* and CΔ*sugC* were compared and analyzed through growth curves, biochemical characteristics, hemolysis characteristics, cell infection tests and pathogenicity tests on BALB/c mice. The results of the growth characteristic experiments in vitro showed that the plateau stage growth period of Δ*sugC* was delayed compared to the TJS75 strain, but there was no difference in the total number of bacteria. The biochemical characteristics and hemolysis ability of Δ*sugC* in sheep blood had no difference compared with TJS75, but its adhesion and invasion abilities in PK-15 cells were decreased. Knockout of the *sugC* gene had no impact on the expression levels of adhesion-related genes in TJS75 in real-time PCR analysis. In addition, the LD_50_ of Δ*sugC* in BALB/c mice was 1.47 × 10^8^ CFU, seven times higher than that of TJS75 (LD_50_ = 2.15 × 10^7^ CFU). These results illustrate that the deletion of *sugC* reduced the virulence of TJS75 to BALB/c mice, but its role in the adhesion and invasion of PK-15 cells in this strain needs to be further explored. In summary, this study provides evidence that the *sugC* gene is a virulence-related gene in the *S. suis* serotype 2 strain and plays a crucial role in the adhesion and invasion of *S. suis*. This study lays a foundation for the further exploration of the potential virulence factors and pathogenesis of *S. suis*.

## 1. Introduction

*Streptococcus suis* (*S. suis*) can infect pig herds and trigger serious diseases such as meningitis, sepsis, arthritis, pneumonia and endocarditis [1]. This bacterium is usually a co-infection agent with other pathogens that cause respiratory diseases in pigs, and it can cause high mortality of up to 80% and result in severe economic losses in the swine industry [2]. Between 1998 and 2005, incidents of *S. suis* serotype 2 infections in human cases and death were reported in China [3]. To date, a total of 29 serotypes of *S. suis* have been reported worldwide, and serotype 2 is the most prevalent and highly pathogenic to humans and pigs. At present, many research studies have focused on analyzing the pathogenic mechanism of *S. suis* and exploring the role of virulence-related factors in the infection processes of highly pathogenic strains [4,5,6]. Research findings have revealed that more than 100 virulence-related factors are associated with the virulence of *S. suis*, including muramidase-released protein (MRP), suilysin (SLY), extracellular protein factor (EPF), fibrinogen-binding protein (FBPS), glutamate dehydrogenase (GDH), phosphate-3-glyceraldehyde dehydrogenase (GAPDH) and capsular biosynthesis (CPS) [7,8,9,10]. However, due to the wide variety of virulence factors and complex modes of action, we still lack a comprehensive understanding of the specific virulence factors and their pathogenesis in *S. suis*.

ATP-binding cassette (ABC) transporters are a type of membrane-integrated protein commonly existing in prokaryotic and eukaryotic organisms and are one of the largest protein families in organisms [11,12]. Various cellular processes, such as the acquisition of essential nutrients, adhesion, conjugation, biofilm formation, toxin secretion and multidrug resistance (MDR), are mediated by ABC transporters [13]. In addition, ABC transporters have been confirmed to be important for the virulence and pathogenesis of *Escherichia coli* (*E. coli*), *Staphylococcus aureus* (*S. aureus*) and *Streptococcus pneumoniae* (*S. pneumoniae*) through mutagenesis studies [14,15]. A *sugC* gene coding for an ABC transporter-related protein was reported to exist in highly pathogenic strains of *S. suis* [15], but its effect on the virulence of *S. suis* remains unknown.

In this study, a wild-type *S. suis* TJS75 strain was used as the parent strain, and a *sugC* gene knockout strain (Δ*sugC*) was successfully constructed by gene homologous recombination technology. The differences in the pathogenicity-related characteristics between Δ*sugC* and the TJS75 strain were compared and analyzed, laying the foundation for further research on the effect of *sugC* in *S. suis*.

## 2. Materials and Methods

### 2.1. Bacterial Strains, Plasmids and Culture Conditions

The *S. suis* TJS75 strain was identified and preserved in the Institute of Animal Husbandry and Veterinary Medicine, Tianjin Academy of Agricultural Sciences. The TJS75 strain and Δ*sugC* strain were cultured in tryptone soy broth (TSB; Solarbio Science & Technology Co., Ltd., Beijing, China) containing 10% fetal bovine serum, and the CΔ*sugC* strain was cultured in TSB (containing 10% fetal bovine serum and an appropriate concentration of spectinomycin). The *E. coli* DH5α was cultured in Luria–Bertani broth (LB; Solarbio Science & Technology Co., Ltd., Beijing, China). Spectinomycin (Spc; Solarbio Science & Technology Co., Ltd., Beijing, China) was added for *S. suis* (100 μg/mL) and *E. coli* (50 μg/mL). Table 1 shows the strains and plasmids used in this study.

### 2.2. Construction of sugC Gene Knockout Strain and Complementary Strain

A *sugC* gene-deficient strain was constructed using homologous recombination technology [17]. The primers used in the experiment are listed in Table 1. Briefly, by using the genomic DNA of the TJS75 strain as a template, the upstream and downstream homologous arms of *sugC* were amplified using primers L1/L2 and R1/R2, respectively. These fragments were connected to the pSET4s plasmid using a DNA Ligation Kit (D6022 DNA Ligation Kit Ver. 2.0.; *TaKaRa* Biomedical Technology (Beijing) Co., Ltd., Beijing, China) to construct the vector pSET4s-Δ*sugC*. Thereafter, pSET4s-Δ*sugC* was electro-transformed into the TJS75 strain [18]. The screening of strains lacking the *sugC* gene was performed using tryptic soy agar (TSA; Solarbio Science & Technology Co., Ltd., Beijing, China) plates and tryptic soy broth, which were contaminated with spectinomycin (Solarbio Science & Technology Co., Ltd., Beijing, China) resistance. Suspected positive strains were identified with primers L1/R2 and S1/S2, and the PCR products were entrusted to **BGI** TECH SOLUTIONS (BEIJING LIUHE) CO., Beijing, China, LIMITED for nucleotide sequencing. The obtained sequences were analyzed and identified using DNAStar5.0.

The sequence fragments of *sugC* and the promoter were amplified with primers CΔ*sugC*-F and CΔ*sugC*-R (Table 1) and cloned into the shuttle plasmid pSET2 (Hangzhou BIO SCI Biotechnology Co., Ltd., Zhejiang, China) of *E. coli* and *S. suis*. Procedures reported in previous studies [7] were used to obtain complement strain Δ*sugC* through cultivation methods and for the identification of suspected positive strains using primers L1/R2 and S1/S2.

In addition, the method of relative quantitative PCR [19] was used to detect the relative expression levels of the *sugC* gene and internal reference gene (*16S rRNA* gene) in the TJS75, Δ*sugC* and CΔ*sugC* strains by q*sugC*-F/q*sugC*-R and q*16S rRNA*-F/q*16S rRNA*-R.

### 2.3. Cultivation Characteristics

Referring to reported studies [20], the morphology and staining characteristics of the TJS75, Δ*sugC* and CΔ*sugC* strains cultured in vitro were observed through Gram staining and microscopy. Briefly, (1) the TJS75, Δ*sugC* and CΔ*sugC* strains were inoculated into a 4.5 mL TSB tube containing 10% fetal bovine serum, and the tube was incubated at 37 °C for 24 h; (2) the 24 h culture medium of TJS75, Δ*sugC* and CΔ*sugC* was used to inoculate TSA plates containing 10% fetal bovine serum; the plates were incubated at 37 °C for 24 h; and the colony characteristics and morphology were observed; (3) Gram staining was conducted to observe the characteristics and microscopic morphologies of the TJS75, Δ*sugC* and CΔ*sugC* strains.

### 2.4. Growth Curve

A sample of the overnight cultures of the TJS75, Δ*sugC* and CΔ*sugC* strains was diluted at a ratio of 1:1000, inoculated into TSB containing 10% fetal bovine serum, and cultured on a shaking platform at 37 °C for 24 h. During the 24 h culture period, 1 mL culture medium was collected every 2 h to inoculate TSA plates containing 10% fetal bovine serum to count the number of growth colonies.

### 2.5. Biochemical Characteristics

A sample of the overnight cultures of the TJS75, Δ*sugC* and CΔ*sugC* strains was inoculated into fermentation broth tubes (Qingdao Hi-Tech Industrial Park Hope Bio-Technology Co., Ltd., Qingdao, China), including xylose, lactose, glucosidase, mannitol, galactose, maltose, sucrose, glycerol, salicin, sorbitol and trehalose. Under sufficient lighting conditions, the reaction tubes and the blank control group were compared and the color changes of the reaction tubes were observed [21]. 

### 2.6. Hemolytic Characteristics

Samples of the overnight cultures of the TJS75, Δ*sugC* and CΔ*sugC* strains were inoculated onto Columbia blood agar plates (Qingdao Hi-Tech Industrial Park Hope Bio-Technology Co., Ltd., Qingdao, China), and incubated in a 37 °C incubator for 24 h to observe and compare the hemolytic characteristics of the gene-deficient and wild strains. Hemolytic characteristics were tested as per a published procedure [22]. Briefly, TJS75, Δ*sugC* and CΔ*sugC* were mixed with a concentration of 50 mL/L defibrated sheep whole blood (Beijing Solarbio Science & Technology Co., Ltd., Beijing, China)–RPMI 1640 medium (Beijing BOAOtoda Technology Co., Ltd., Beijing, China), centrifugated at 2000 rpm/min for 2 min, and then incubated at 37 °C. A total of 4 groups in an hourly interval (i.e., 0 h, 1 h, 2 h, and 3 h) were set, and each group was set with a negative control. When each group of tubes was taken out, they were centrifugated at 12,000 rpm/min for 1 min to collect the supernatant and measure the OD_630_ values. 

### 2.7. Characteristics of Adhesion and Invasion

#### 2.7.1. Exploring the Minimum Bactericidal Concentration of GEN for TJS75, Δ*sugC* and CΔ*sugC *

To prepare for the adhesion and invasion testing, the broth microdilution method was conducted with a 96-well cell culture plate, using *E. coli* ATCC25922 as the quality control strain, to verify the effectiveness of gentamicin (GEN; Beijing Solarbio Science & Technology Co., Ltd., Beijing, China) (NEST; Wuxi NEST Biotechnology Co., Ltd., Wuxi, China) [23]. Subsequently, this method was applied to determine the minimum inhibitory concentrations (MICs) of TJS75, Δ*sugC* and CΔ*sugC*. According to the MIC measurement results, the appropriate concentration of GEN was interacted with the test strains (1.0 × 10^8^ CFU/mL) for 1 h, and the test strain solution was recorded as 10^0^ at this time. A volume of 100 μL of the bacterial solution was added to 900 μL TSB, vortexed for 30 s, and denoted as the 10**^−^**^1^ bacterial solution. According to this method, 10**^−^**^2^, 10**^−^**^3^, 10**^−^**^4^, 10**^−^**^5^, 10**^−^**^6^ and 10**^−^**^7^ bacterial solutions were obtained sequentially. The 100 μL bacterial solution with different dilution ratios was coated on TSA plates and incubated at 37 °C to observe whether there was bacterial growth on the plates. The minimum bactericidal concentration (MBC) of GEN at 1 h against TJS75, Δ*sugC* and CΔ*sugC* was explored.

#### 2.7.2. The Effect of GEN on the Morphology of PK-15 Cells

A concentration of 1×MBC of GEN at 1 h was mixed with PK-15 cells (1.0 × 10^9^/L), and they were incubated at 37 °C for 1 h to observe the cell state. At the same time, the supernatant was removed and cell growth medium (MEM containing 10% fetal bovine serum) was added to PK-15 cells for 48 h to explore the effect of the appropriate concentration of GEN by MTT [24].

#### 2.7.3. Bacterial Adherence and Invasion Assays

The adhesion and invasion abilities of TJS75, Δ*sugC* and CΔ*sugC* on PK-15 cells were analyzed in accordance with the published methods [20]. In brief, the bacterial suspensions of TJS75, Δ*sugC* and CΔ*sugC* were added into a 24-well plate pre-cultured with a monolayer of PK-15 cells (approximately 1.0 × 10^9^/L), and they were inoculated at an MOI of 100:1. Each strain was divided into 6 groups and incubated at 37 °C for 3 h. Randomly, 3 groups were selected and washed with sterile PBS (Beijing Solarbio Science & Technology Co., Ltd., Beijing, China) 6 times, and then 1 mL of sterile distilled water was added for digestion for 10 min. The above liquid was applied to the TSA plates, and the cells were grown for 24 h to count the number of colonies. The result was recorded as the number of cell adhesion bacteria in 3 h. Then, the remaining 3 parallel groups were treated with GEN (final concentration 100 μg/mL) for 1 h. The supernatant was removed, the pallet was washed with PBS 6 times, and we then added 1 mL of sterile distilled water for digestion for 10 min. Then, the above liquid was applied to the TSA plates, and the cells were grown for 24 h to count the number of colonies. The counting result was the number of viable intracellular bacteria grown in 3 h. 

#### 2.7.4. The Relative Expression of *fbps*, *cps2J*, *gdh* and *gapdh*

A real-time PCR [19] was used to detect the relative expression of *fbps*, *cps2J*, *gdh* and *gapdh*. The raw data were analyzed using *16S rRNA* as an internal reference gene. The primers used for this analysis are listed in Table 1.

### 2.8. Animal Pathogenicity Tests

The tested strains were grown overnight or for 24 h at 37 °C and then diluted with 0.9% NaCl to 1 × 10^9^ CFU/mL, 1 × 10^8^ CFU/mL, 1 × 10^7^ CFU/mL and 1 × 10^6^ CFU/mL. BALB/c mice (4-week-old females) (SPF (Beijing) Biotechnology Co., Ltd., Beijing, China) were intraperitoneally injected with the test strains of the bacterial solution [25]. According to the different doses of the tested strains, the mice were randomly divided into 4 groups (three study groups with 40 mice in each group and one control group with 10 mice) (Table 2). All 4 groups of experimental mice were fed in IVC mouse cages and observed every 12 h for clinical manifestations and mortality within the trial period of 7 days. The median lethal dose (LD_50_) was calculated according to the deaths of the BALB/c mice. In addition, surviving mice were deeply anesthetized and euthanized through cervical dislocation. The death of a mouse was considered when it was approaching death and its main symptoms included coarse and messy fur, loss of appetite, weight loss, increased secretion from the corners of the eyes, a hunched back, and lying still.

### 2.9. Statistical Analysis

All experimental data obtained in this study were analyzed using GraphPad 5.0 and represented as the average and standard deviation of three experimental results. Two-way analysis of variance (ANOVA) was used for the data; *p* < 0.05 was considered significant difference, *p* < 0.01 was considered very significant difference and *p* < 0.001 was considered extremely significant difference.

## 3. Results

### 3.1. Identification of sugC Gene Knockout Strain and Complementary Strain

The primers listed in Table 1 were used to verify the deletion through PCR. Some fragments were amplified using primers L1/R2 in Δ*sugC*. The fragments in Δ*sugC* were smaller than those from the parent strain TJS75 (Figure 1 and Figure 2; Appendix A). The *sugC* expression was detected by qRT-PCR with primers q*sugC*-F/q*sugC*-R. The results showed that the expression of the *sugC* gene could be detected in TJS75 and CΔ*sugC*, and there was no significant difference in the expression level (Figure 3), but it was not found in Δ*sugC*. These results indicate that we constructed a *sugC* gene knockout strain (Δ*sugC*) and its complementary strain (CΔ*sugC*). The original gel images of Figure 1 can be found in the Appendix A.

### 3.2. Observation of the Culture Characteristics of S. suis Strains

When Δ*sugC*, CΔ*sugC* and TJS75 were inoculated in TSB containing 10% fetal bovine serum, Δ*sugC* reached a logarithmic growth phase after 8 h cultivation and reached its peak at 16 h; afterwards, its growth tended to flatten out. The CΔ*sugC* and TJS75 strains reached a logarithmic growth phase at 8 h cultivation, reached a peak about 14 h, and then tended to flatten out. Although the growth rate of Δ*sugC* decreased compared to TJS75 and CΔ*sugC*, there were no significant differences from 14 h to 24 h (Figure 4) in the total bacterial counts, which indicated that the time for Δ*sugC* to reach the plateau stage was relatively delayed, but the effect of the bacterial content after reaching the peak did not have a significant difference. Gram staining and microscopic examination showed a similar morphology to TJS75 and CΔ*sugC*, and all strains were Gram-positive, with single or multiple short-chain arranged cocci.

### 3.3. Effect of the sugC Gene on the Biochemical Characteristics of the TJS75 Strain

Δ*sugC*, CΔ*sugC* and TJS75 could ferment α-glucosidase, galactose, maltose, sucrose, salicin and trehalose, but could not ferment sorbitol, mannitol, xylose, lactose and glycerol. These results indicate that the deletion of the *sugC* gene had no significant influence on the biochemical characteristics in TJS75.

### 3.4. Lacking the sugC Gene Does Not Change the Hemolytic Characteristics of the TJS75 Strain

The semi-transparent and round colonies of Δ*sugC* on the Columbia blood agar plate were accompanied by α-hemolysis, and this result was consistent with CΔ*sugC* and TJS75. At the same time, Δ*sugC*, CΔ*sugC* and TJS75 were interacted with sheep red blood cells, and the average OD_630_ values of the treated solution were 0.3050, 0.3110 and 0.3180, respectively. The difference for Δ*sugC* was not significant compared to CΔ*sugC* and TJS75 (Figure 5), indicating that the deletion of *sugC* did not significantly affect the hemolytic ability of TJS75 in sheep red blood cells.

### 3.5. Knocking Out the sugC Gene Reduces the Adhesion and Invasiveness of TJS75 in PK-15 Cells

The MICs of GEN (Appendix A) against Δ*sugC*, CΔ*sugC* and TJS75 were determined to be 16 μg/mL (Appendix A), and the MBCs for 1 h were 128 μg/mL (Figure 6A). Next, 128 μg/mL GEN was used to interact with PK-15 cells for 1 h, and the cell viability of PK-15 cells (Figure 6B) did not significantly change compared with the control group.

The results of the cell adhesion tests showed that the number of viable bacteria on the surface and in the interior of PK-15 cells of Δ*sugC* was significantly reduced compared to TJS75 and CΔ*sugC* showed no significant change compared to TJS75 (Figure 6C). After using GEN to inactivate the living bacteria adhering to the surfaces of PK-15 cells, the number of Δ*sugC* living bacteria inside PK-15 cells decreased compared to TJS75 and CΔ*sugC* (Figure 6D). After analyzing the expression levels of genes mainly involved in adhesion-related virulence in TJS75, Δ*sugC* and CΔ*sugC*, it was found that the deletion of the sugC gene had no significant effect on the mRNA expression levels of genes such as *fbps*, *cps2J*, *gdh* and *gapdh* (Figure 6E). These results indicate that the deletion of *sugC* reduced the adhesion and invasion abilities of TJS75 in PK-15 cells.

### 3.6. Deletion of the sugC Gene Reduced the Virulence of TJS75 in BALB/c Mice

After inoculation, the three study groups of BALB/c mice showed varying degrees of rough and matte fur, decreased appetite, weight loss, mental depression, and increased secretion from the corners of the eyes. The control group of mice treated with 0.9% NaCl showed no abnormalities. At the dose of 1.0 × 10^9^ CFU/mL, all BALB/c mice attacked with TJS75 began to develop the disease within 12 h and all died within 24 h, as seen for mice challenged with Δ*sugC* and CΔ*sugC*. At the dose of 1.0 × 10^8^ CFU/mL, BALB/c mice infected with TJS75 began to develop the disease at 12 h (the incidence rate was 100% at 24 h) and died within 24 h (the death rate was 60% at the end of the experiment). BALB/c mice infected with CΔ*sugC* were similar to those with TJS75, but the death rate was 70% at the end of the experiment. Meanwhile, BALB/c mice infected with Δ*sugC* were also similar to those with TJS75 and CΔ*sugC* in the time of onset, but the incidence rate, time of death and mortality rate were 36 h, 80% and 40%, respectively, which were different from those of TJS75 and CΔ*sugC*. At the doses of 1.0×10^7^ CFU/mL and 1.0 × 10^6^ CFU/mL, the times of onset (24 h) in all BALB/c mice infected with the three tested strains were consistent, but the morbidity rates and mortality rates of CΔ*sugC* and TJS75 were higher than those of Δ*sugC* (Table 3 and Table 4). According to the Reed–Munch method, the LD_50_ of each group was calculated, and the results showed that the LD_50_ values of Δ*sugC*, CΔ*sugC* and TJS75 were 1.47 × 10^8^ CFU, 1.75 × 10^7^ CFU and 2.15 × 10^7^ CFU (Table 4). Interestingly, the LD_50_ of Δ*sugC* in BALB/c mice was seven times higher than that of TJS75, and these results indicate that the deletion of *sugC* reduced the virulence of TJS75 in BALB/c mice. In addition, the distribution of bacteria in the organs showed that the three tested strains could be isolated from the brains, hearts, livers, spleens, lungs and kidneys of dead mice in each experimental group. After 48–96 h of infection, varying amounts of the three tested strains could be isolated from the brains, heart blood and lungs of dying mice. It should be noted that a small amount of bacteria could be isolated from surviving mice with infection at 7 days (Table 5). At the same time, the number of dead mice (including dead and near dead) in Δ*sugC* was lower than that in TJS75 and CΔ*sugC*, and the number of surviving mice was higher than that in the TJS75 and CΔ*sugC* groups.

## 4. Discussion

This study used homologous recombination technology to exchange the upstream and downstream homologous fragments of the *sugC* gene with the genome of the *S. suis* wild strain TJS75, thereby replacing the *sugC* gene fragments and obtaining the target gene deletion strain. Then, the TJS75 strain was used as the research object to explore the main functions of the knockout *sugC* gene. The above method was adopted as the main principle of target gene deletion and mutation [16]. In order to reveal the virulence relationship between the *sugC* gene and wild-type *S. suis* TJS75, the construction of *sugC* gene mutant strains was performed to explore the function of virulence-related factors, and this method is widely used [7,17,18,20,25,26].

ABC transporters are a family of membrane proteins that have multiple functions and are widely distributed in bacteria [27]. Their main function is to utilize the energy generated by ATP hydrolysis to achieve the transmembrane transport of substrates [28]. Most ABC transporters were initially discovered through the study of drug resistance in eukaryotic organisms [29]. There is currently extensive research on the role of ABC transporters in bacterial pathogenicity. For example, during the process of human infection with *Streptococcus agalactiae* CS101, the Opp (oligopeptide) ABC transporters, which are involved in amino acid intake, heme synthesis, spore formation, and the expression of the major virulence factor SpeB cysteine protease, were observed to reduce the expression of fibrinogen-binding protein (FbsA), indicating a decrease in the adhesion of the CS101 strain to epithelial cells [30]. Research has shown that *Fusarium graminearum,* which is a destructive fungal pathogen of small grain crops worldwide, contains an ABC transporter protein (called FgAtm1) involved in regulating iron homeostasis (which is important for the growth, reproduction, and other metabolic processes of all eukaryotes). Here, the deficiency of FgAtm1 reduced the activity of Fe-S protein nitrite reductase and xanthine dehydrogenase in the cytoplasm, thereby activating the transcription factor FgAreA and leading to the high expression of FgHopX, which directly inhibits the transcription of iron-consuming protein genes [31]. The ABC transporter encoded by the *sugC* gene in *Mycobacterium* could provide trehalose for *Mycobacterium* and enhance its adaptability to the environment [32,33]. The survival time of mice infected with the *sugC* gene deletion strain (Δ*sugC*) was 33 days, which was longer than the survival time of mice infected with the wild strain; this indicates that the *sugC* gene enhanced the virulence of *Mycobacteria* and is an important virulence factor [32]. Recently, the genes of different serotypes of *S. suis* were examined through comparative genomics, and the results showed that the *sugC* gene existed in highly virulent *S. suis* [14], but these studies did not explore the correlation between the *sugC* gene and the virulence of *S. suis*. Fortunately, we obtained a TJS75 strain carrying the *sugC* gene and possessing pathogenic abilities from veterinary clinical practice. On the basis of the TJS75 strain, we designed specific primers to knock out and supplement the *sugC* gene, obtaining Δ*sugC* and CΔ*sugC* strains, respectively.

The infection mechanisms of *S. suis* can be divided into adhesion and colonization, blood survival and diffusion, inflammatory activation and septic shock, invasion of the central nervous system and meningitis, among which adhesion to the host tissue or cells is the most important step in the invasion and infection of host cells [34]. At present, CPS, MRP, GDH, FBPS and GAPDH are known to be the adhesins of *S. suis* serotype 2, which are involved in the adhesion of *S. suis* to host tissue or cells and enhance the pathogenicity to the host [35,36,37,38]. The absence of one or more adhesins could reduce the adhesion ability of *S. suis* serotype 2 to host tissue or cells, but it would not be completely lost, indicating the presence of other adhesins. The growth and biochemical characteristics, hemolysis, adhesion, invasiveness, and pathogenicity to mice were evaluated between Δ*sugC* and its parent strain. Interestingly, we found that there was no significant difference in the growth characteristics, biochemical characteristics, and hemolysis ability of Δ*sugC* compared to the wild strain TJS75, but the ability to adhere and invade PK-15 cells was decreased. At the same time, we found that there were no obvious impact on the relative expression levels of adhesion-related genes such as *fbps*, *gdh*, *gapdh* and *cps2J* between Δ*sugC* and its parent strain TJS75. This means that the deletion of the *sugC* gene would not affect the expression levels of the *fbps*, *gdh*, *gapdh* and *cps2J* genes in the TJS75 strain. It could be seen that the deletion of the *sugC* gene could reduce the adhesion and invasion abilities of TJS75 in PK-15 cells, but it was not completely blocked. Meanwhile, the results of the animal pathogenicity test also showed that the LD_50_ of Δ*sugC* in BALB/c mice was 1.47 × 10^8^ CFU, and there was a seven-fold difference compared to the TJS75 strain (LD_50_ = 2.15 × 10^7^ CFU), indicating that the virulence of TJS75 to BALB/c mice was reduced by the deletion of the *sugC* gene. Thus, the above results indicate that the *sugC* gene is a virulence-related gene in *S. suis* serotype 2.

## 5. Conclusions

In this study, a wild-type strain of TJS75 isolated from fattening pigs’ brain tissue samples was used as a parent strain, and a knockout *sugC* strain (Δ*sugC*) and complementary strain (CΔ*sugC*) were successfully constructed using homologous recombination technology. Through experimental analysis, it has been confirmed that the *sugC* gene is a new virulence-related gene in the TJS75 strain both in vivo and in vitro. However, the specific mechanism of action of this gene remains to be explored. This study lays a foundation for the further exploration of the potential virulence factors and pathogenesis of *S. suis*.

## Figures and Tables

**Figure 1 vetsci-11-00447-f001:**
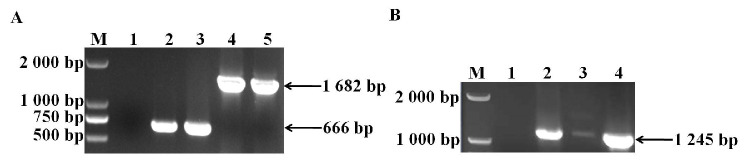
Construction and confirmation of Δ*sugC* and CΔ*sugC*. (**A**) Construction and confirmation of Δ*sugC.* M = DL2000 marker; 1 = ddH_2_O; 2 and 4 = randomly chosen clone strains; 3 = pSET4s-Δ*sugC*; 5 = TJS75. (**B**) Construction and confirmation of CΔ*sugC.* M = DL2000 marker; 1 = ddH_2_O; 2 = randomly chosen clone strain; 3 = pSET2-*sugC*; 4 = TJS75.

**Figure 2 vetsci-11-00447-f002:**
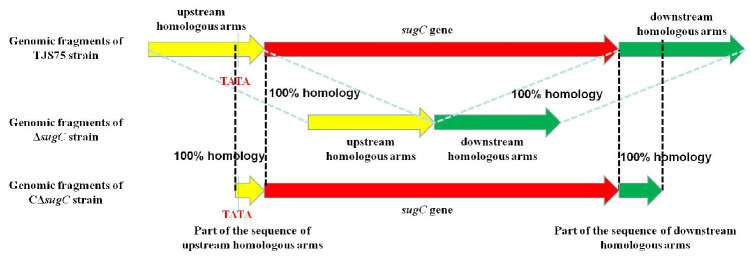
Amino acid sequence analysis of PCR products from multiple target fragments with TJS75, Δ*sugC* and CΔ*sugC*.

**Figure 3 vetsci-11-00447-f003:**
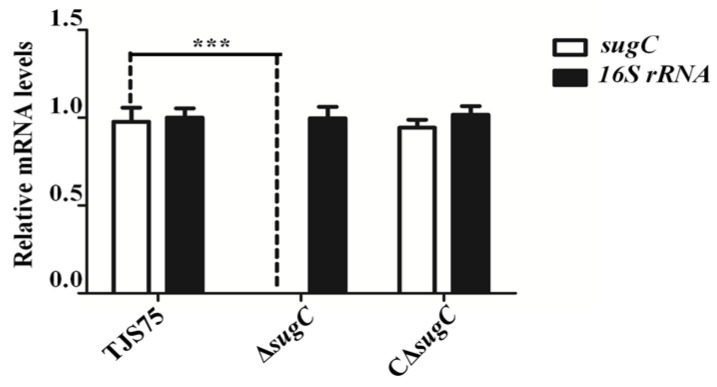
Transcription of target molecules in Δ*sugC*, CΔ*sugC* and TJS75. Δ*sugC* vs. TJS75: *p_sugC_* = 0, *p_16S rRNA_* = 0.8839; CΔ*sugC* vs. TJS75: *p_sugC_* = 0.7962, *p_16S rRNA_* = 0.6983. ***: *p* < 0.001.

**Figure 4 vetsci-11-00447-f004:**
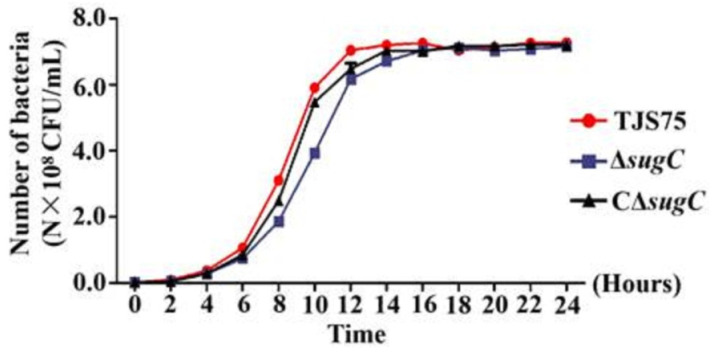
In vitro growth curves of different generations of TJS75, Δ*sugC* and CΔ*sugC*. Δ*sugC* vs. TJS75: *p*_6h_ = 0.0064, *p*_8h_ = 0.0035, *p*_10h_ = 0.0051, *p*_12h_ = 0.0063, *p*_14h_ = 0.0912, *p*_16h_ = 0.2954, *p*_20h_ = 0.2149, *p*_24h_ = 0.0506; CΔ*sugC* vs. TJS75: *p*_6h_ = 0.1460; *p*_8h_ = 0.0116; *p*_10h_ = 0.0092, *p*_12h_ = 0.1077, *p*_14h_ = 0.2254, *p*_16h_ = 0.0965, *p*_20h_ = 0.5040, *p*_24h_ = 0.1165.

**Figure 5 vetsci-11-00447-f005:**
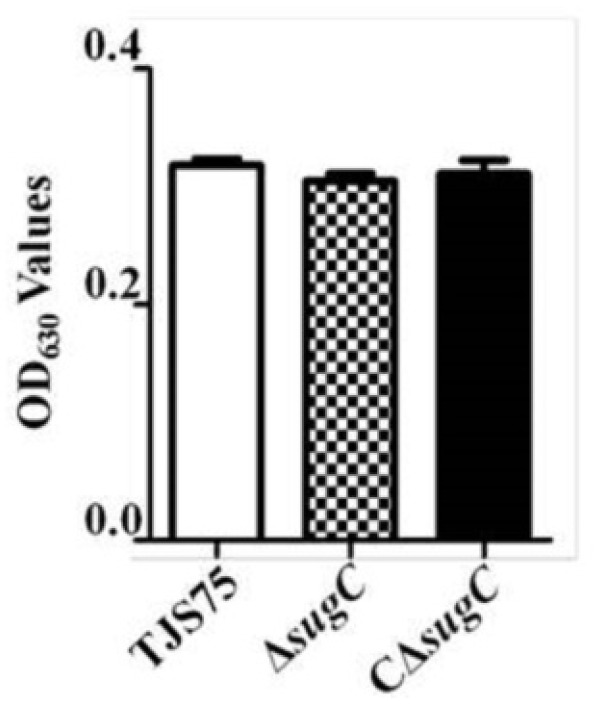
Hemolytic characteristic results of detection of *sugC* in *S. suis*. OD_630_ values of solutions in which Δ*sugC*, CΔ*sugC* and TJS75 were interacted with sheep red blood cells. Mean_TJS75_ = 0.3180, Mean_Δ*sugC*_ = 0.3050, Mean_CΔ*sugC*_ = 0.3110, *p*_Δ*sugC*_ vs. _TJS75_ = 0.1421, *p*_C__Δ*sugC*_ vs. _TJS75_ = 0.3714.

**Figure 6 vetsci-11-00447-f006:**
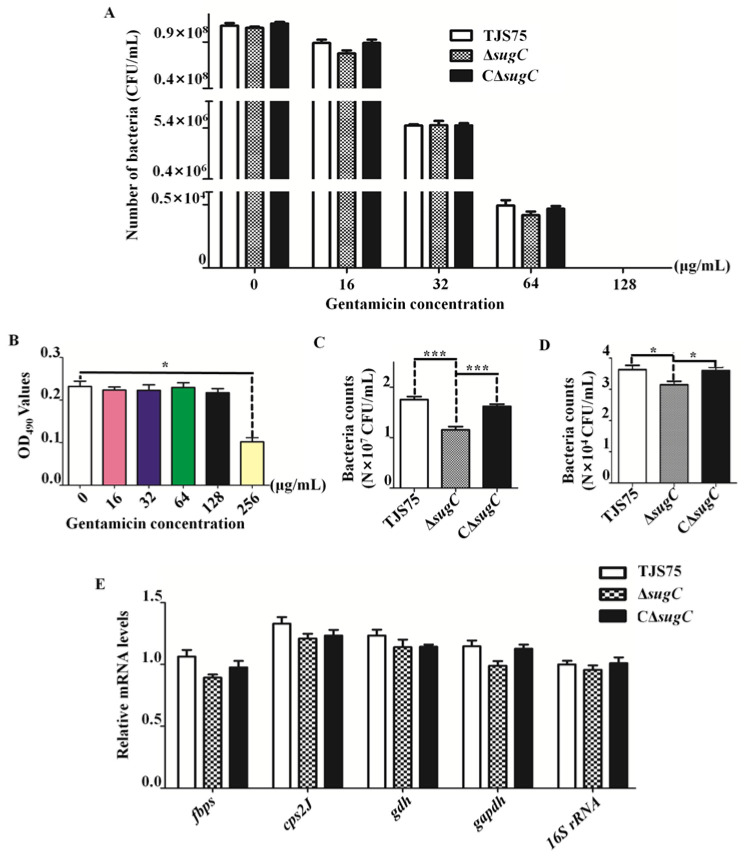
Analysis of adhesion and invasiveness. (**A**) The MBCs for 1 h of Δ*sugC*, CΔ*sugC*, and TJS75. (**B**) The effect of GEN on the viability of PK-15 cells by MTT. Mean_0 μg/mL_ = 0.2320, Mean_16 μg/mL_ = 0.2224, Mean_32 μg/mL_ = 0.2229, Mean_64 μg/mL_ = 0.2296, Mean_128 μg/mL_ = 0.2173, Mean_256 μg/mL_ = 0.1021; *p*_0 μg/mL_ vs. _16 μg/mL_= 0.5014, *p*_0 μg/mL_ vs. _32 μg/mL_= 0.5805, *p*_0 μg/mL_ vs. _64 μg/mL_ = 0.8622, *p*_0 μg/mL_ vs. _128 μg/mL_ = 0.1952, *p*_0 μg/mL_ vs. _256 μg/mL_= 0.0102. (**C**) Results of cell adhesion test of Δ*sugC*, CΔ*sugC* and TJS75. Mean_TJS75_ = 1.76 × 10^7^ CFU/mL, Mean_Δ*sugC*_ = 1.16 × 10^7^ CFU/mL, Mean_CΔ*sugC*_ = 1.62 × 10^7^ CFU/mL, *p*_Δ*sugC*_ vs. _TJS75_ = 0.0001, *p_C_*_Δ*sugC*_ vs. _TJS75_ = 0.0918, *p*_Δ*sugC*_ vs. _CΔ*sugC*_ = 0.0001. (**D**) Results of cell invasion test of Δ*sugC*, CΔ*sugC*, and TJS75. Mean_TJS75_ = 3.63 × 10^4^ CFU/mL, Mean_Δ*sugC*_= 3.14 × 10^4^ CFU/mL, Mean_CΔ*sugC*_ = 3.60 × 10^4^ CFU/mL, *p*_Δ*sugC*_ vs. _TJS75_ = 0.0307, *p_C_*_Δ*sugC*_ vs. _TJS75_ = 0.8327, *p*_Δ*sugC*_ vs. _CΔ*sugC*_ = 0.0165. (**E**) The relative expression of *fbps*, *cps2J*, *gdh* and *gapdh*. Δ*sugC* vs. TJS75: *p_fbps_* = 0.0624, *p cps2J* = 0.1124, *p gdh* = 0.2038, *p gapdh* = 0.0528, *p_16S rRNA_* = 0.6622; CΔ*sugC* vs. TJS75: *p_fbps_* = 0.8320, *p cps2J* = 0.2907, *p gdh* = 0.2851, *p gapdh* = 0.6814, *p_16S rRNA_* = 0.8105. *: *p* < 0.05; ***: *p* < 0.001.

**Table 1 vetsci-11-00447-t001:** Information about strains, plasmids and primers used in this study.

Type and Name	Sequence (5′-3′)	Purpose
TJS75	/	*S. suis* serotype 2 virulence strain isolated from diseased pigs in Tianjin in 2015 (Accession: CP095162.1)
Δ*sugC*	/	TJS75 strain with *sugC* gene knocked out
CΔ*sugC*	/	Complementary strain of Δ*sugC*; *Spc*^R^
*E. coli* DH5α	/	For cloning of recombinant plasmids
pSET4s [16]	/	*S. suis* temperature-sensitive suicide vector
pSET2	/	*E. coli*–*S. suis* shuttle vector; *Spc*^R^
pSET4s-Δ*sugC*	/	Recombinant vector with background of pSET4s, designed to knock out *sugC* gene; *Spc*^R^
pSET2-*sugC*	/	pSET2 containing complete *sugC* gene and its promoter; *Spc*^R^
S1	TACTACTTACCTCCGTATTGCA	Detecting full length of *sugC* gene
S2	TGATTACCTTTAACGATAT	
L1	GAAG**CTGCAG**TCAAAGAAGACATATACCCAAG ^1^	Detecting upstream homologous arms of *sugC* gene
L2	GAGGTGTGATTGCTCAAAGATAT	
R1	TAGCCACGTTACACACCTC	Detecting downstream homologous arms of *sugC* gene
R2	CCC**CCCGGG**CGAAGCTGAACGTGGCTAT	
CΔ*sugC*-F	C**CCCGGG**TATATGATGAAGGCTACCAGCAACCACA	*sugC* gene and its upstream promoter carrying relevant restriction enzyme sites at both ends
CΔ*sugC*-R	G**CTGCAG**GAAATTAAAGACTTTGCAAGCAGCGT	
q*sugC*-F	CTACTTACCTCCGTATTGCATAATG	Relative quantitative detection of *sugC* gene
q*sugC*-R	CCATGTTATTGATGATGTCGTGACT	
q*16S rRNA*-F	GGCGTGCCTAATACATG	Relative quantitative detection of internal reference genes
q*16S rRNA*-R	GCTATGAGGCAGGTT	
q*gdh*-F	CGGCGGTGGTAAAGGTGGTT	Relative quantitative detection of *gdh* gene
q*gdh*-R	CGTCAAGTGAAGGTCCGATGTG	
q*fbps*-F	TGCCATTTGCCAATAGCCCTGAA	Relative quantitative detection of *fbps* gene
q*fbps*-R	TCCCGCTCCGCCTTATCCTG	
q*cps2J*-F	GTTACTTGCTACTTTTGATGG	Relative quantitative detection of *cps2J* gene
q*cps2J*-R	TTTTCATTTCCTAAGTCTCG	
q*gapdh*-F	GTTTGATGACTACAATCCTCGGTTAC	Relative quantitative detection of *gapdh* gene
q*gapdh*-R	GCTTTAGCAGCACCAGTTGAG	

^1.^ Underlined and bold type indicates the restriction sites in the primer sequences.

**Table 2 vetsci-11-00447-t002:** Test group.

Dose (CFU/mL)	Strain and Number of Mice			
	TJS75	Δ*sugC*	CΔ*sugC*	Control (0.9% NaCl)
1 × 10^9^	10	10	10	/ ^1^
1 × 10^8^	10	10	10	/
1 × 10^7^	10	10	10	/
1 × 10^6^	10	10	10	/
Total	40	40	40	10

^1^ “/” means that there was no such content.

**Table 3 vetsci-11-00447-t003:** The initial times of onset and death and the number of infected mice in each group.

Dose (CFU/mL)	Initial Time of Onset (h)			Initial Time of Death (h)			Duration of Onset (h)			Morbidity Rate %		
	TJS75	Δ*sugC*	CΔ*sugC*	TJS75	Δ*sugC*	CΔ*sugC*	TJS75	Δ*sugC*	CΔ*sugC*	TJS75	Δ*sugC*	CΔ*sugC*
1.0 × 10^9^	12	12	12	12	12	12	12	12	12	100	100	100
1.0 × 10^8^	12	12	12	24	24	24	132	132	132	100	80	100
1.0 × 10^7^	12	24	12	48	0	24	132	132	132	80	40	80
1.0 × 10^6^	24	24	24	60	0	60	24	12	36	60	20	60

**Table 4 vetsci-11-00447-t004:** The LD_50_ of Δ*sugC*, CΔ*sugC* and TJS75 in BALB/c mice.

Dose (CFU/mL)	Number of Dead Mice/Total Mice								
	TJS75			Δ*sugC*			CΔ*sugC*		
1.0 × 10^9^	10/10	10/10	10/10	10/10	10/10	10/10	10/10	10/10	10/10
1.0 × 10^8^	6/10	7/10	5/10	4/10	4/10	4/10	7/10	7/10	7/10
1.0 × 10^7^	4/10	4/10	4/10	0/10	0/10	0/10	4/10	4/10	4/10
1.0 × 10^6^	2/10	2/10	2/10	0/10	0/10	0/10	2/10	2/10	2/10
Mean LD_50_ value	2.15 × 10^7^ CFU			1.47 × 10^8^ CFU			1.75 × 10^7^ CFU		

**Table 5 vetsci-11-00447-t005:** Distribution and bacterial count determination of three tested strains in BALB/c mouse organs.

Group	Dose (CFU/mL)	Dying Mice	Average Bacterial Count			Survivors	Average Bacterial Count		
			Heart Blood	Brain	Lungs		Heart Blood	Brain	Lungs
TJS75	1.0 × 10^8^	4	3.40 × 10^6^	1.10 × 10^5^	5.10 × 10^5^	4	1.10 × 10^3^	1.40×10^1^	3.0 × 10^1^
	1.0 × 10^7^	2	4.90 × 10^5^	5.40 × 10^4^	6.40 × 10^4^	6	4.80 × 10^1^	4	1
	1.0 × 10^6^	1	5.80 × 10^4^	1.30 × 10^2^	8.40 × 10^1^	8	2	0	0
Δ*sugC*	1.0 × 10^8^	1	7.80 × 10^4^	6.30 × 10^2^	3.80 × 10^1^	6	5.20 × 10^2^	9	1.40 × 10^1^
	1.0 × 10^7^	0	/ *	/	/	10	1.50 × 10^1^	0	0
	1.0 × 10^6^	0	/	/	/	10	0	0	0
CΔ*sugC*	1.0 × 10^8^	3	8.60 × 10^6^	7.10 × 10^4^	4.90 × 10^5^	3	3.20 × 10^2^	4	6.70 × 10^1^
	1.0 × 10^7^	1	3.80 × 10^5^	1.70 × 10^3^	4.50 × 10^4^	6	2.60 × 10^1^	4	0
	1.0 × 10^6^	1	5.70 × 10^4^	4.10 × 10^2^	5.90 × 10^1^	8	1	0	0

* “/” means no such result.

## Data Availability

The complete genome sequence of the TJS75 strain (accession: CP095162.1) has been submitted to GenBank.

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
