# Peer review of "Characterization Studies on the sugC Gene of Streptococcus suis Serotype 2 in Adhesion, Invasion, and Virulence in Mice"

_vetsci, 2024, doi:10.3390/vetsci11090447_

Round 1
Reviewer 1 Report
Comments and Suggestions for Authors
The manuscript entitled "Characterization studies on the sugC gene of Streptococcus suis 2 in adhesion, invasion, and virulence in mice" analyzes the primary function of the potential virulence factor sugC in S. suis. The results show that sugC participated in the adhesion and invasion of S. suis to PK-15 cells and affected the virulence of S. suis in mice. But the paper cannot be accepted in its present form. The lists are the major compulsory revisions:
1. In the abstract, the words “specific regulatory mechanism” and “regulating the adhesion and invasion of S. suis” should not appropriate and must be revised.
2. What the Accession number of sugC gene? Please add the information in the manuscript.
3. In the line 85-86, “The TJS75 strain and its mutant were cultured in Tryptone Soy Broth”. As we know, S. suis demands high nutrition, so it can grow in THB or TSB containing 10% serum. Moreover the complemented strain of ΔsugC (CΔsugC) has the plasmid of pSET2-sugC with SpcR , so CΔsugC should grow with Spectinomycin. The author should confirm all the culture medium in the experiments and maintain consistency in the whole manuscript.
4. The real-time PCR were analyzed using 16S rRNA as an internal reference gene. This needs a reference. Why not use other house-keeping genes? Because the ribosome RNA genes have several copies in the bacterial genome, and the mRNA level of 16S rRNA are usually significant higher than other genes.
5. In the line 182-183, the sentence “Applying the above liquid to the TSA plates and growing for 3 h for counting the number of colonies.” should be rewritten.
6. In the line 205-206, “If the weight exceeds 20% of the average weight of the control group mice, the mouse would be considered dead.” Is the judgement mentioned correct? And also needs a reference.
7. In the discussion, the paragraph 2 from line 353-366 should be deleted because the gene knockout method based on pSET4s is very conventional. Meanwhile, the known functions of the ATP-binding cassette transporter encoded by sugC gene in different bacteria could added in this section.
8. Figure A1 is redundant.
Author Response
Comments 1: In the abstract, the words “specific regulatory mechanism” and “regulating the adhesion and invasion of S. suis” should not appropriate and must be revised |
Response 1: Thank you for pointing this out. I agree with this comment. Therefore, In the abstract, the statement containing “specific regulatory mechanism” and “regulating the adhesion and invasion of S. suis” has been modified to “…, but its role in the adhesion and invasion of PK-15 cells by this strain would need to be further explored.”(Page 1,Abstract, line 15-16.)
|
Comments 2: What the Accession number of sugC gene? Please add the information in the manuscript. |
Response 2: Thank you for pointing this out. I have uploaded the whole genome information of the TJS75 strain containing the sugC gene to GenBank and obtained the accession number( CP095162.1).The accession number of the sugC gene was not obtained because this gene was not uploaded separately. If you think the accession number is necessary, I will apply to GenBank.( Page 13, Data Availability Statement, line 6-7.) |
Comments 3: In the line 85-86, “The TJS75 strain and its mutant were cultured in Tryptone Soy Broth”. As we know, S. suis demands high nutrition, so it can grow in THB or TSB containing 10% serum. Moreover the complemented strain of ΔsugC (CΔsugC) has the plasmid of pSET2-sugC with SpcR , so CΔsugC should grow with Spectinomycin. The author should confirm all the culture medium in the experiments and maintain consistency in the whole manuscript. |
Response 3: Thank you for pointing this out. I agree with this comment. Based on your suggestion, I have unified the relevant descriptions in the article.” The TJS75 strain and ΔsugC strain were cultured in Tryptone Soy Broth (TSB; Solarbio Science & Technology Co., Ltd.; Beijing, China) containing 10% fetal bovine serum , and CΔsugC strain was cultured in TSB (containing 10% fetal bovine serum and appropriate concentration of Spectinomycin).”(Page 2, “2.1. Bacterial strains, plasmids and culture conditions”, line 2-6; Page 4, “2.4. Growth curve”, line 4-5.) |
Comments 4: The real-time PCR were analyzed using 16S rRNA as an internal reference gene. This needs a reference. Why not use other house-keeping genes? Because the ribosome RNA genes have several copies in the bacterial genome, and the mRNA level of 16S rRNA are usually significant higher than other genes. |
Response 4: Thank you for pointing this out. Through literature review, I found that the 16S rRNA gene is one of the highly conserved genes in bacterial genomes, and many articles (references A~E) use the 16S rRNA gene as an internal reference gene. Therefore, I also applied this gene as an internal reference gene. Meanwhile, I believes that it is feasible to use the same highly conserved gene as an internal reference gene to evaluate the mRNA level differences of virulence genes between the parent strain and its mutants, regardless of whether the internal reference gene is multi-copy or single copy. [A] Zuo J, Fan Q, Li J, Liu B, Xue B, Zhang X, Yi L, Wang Y. Sub-Inhibitory Concentrations of Amoxicillin and Tylosin Affect the Biofilm Formation and Virulence of Streptococcus suis. Int J Environ Res Public Health. 2022, 19(14), 8359. doi: 10.3390/ijerph19148359. ( References cited by the author) [B] Li S, Zhou Y, Yuan T, Feng Z, Zhang Z, Wu Y, Xie Q, Wang J, Li Q, Deng Z, Yu Y, Yuan X. Selection of internal reference gene for normalization of reverse transcription-quantitative polymerase chain reaction analysis in Mycoplasma hyopneumoniae. Front Vet Sci. 2022, 9, 934907. doi: 10.3389/fvets.2022.934907. [C] Badell E, Guillot S, Tulliez M, Pascal M, Panunzi LG, Rose S, Litt D, Fry NK, Brisse S. Improved quadruplex real-time PCR assay for the diagnosis of diphtheria. J Med Microbiol. 2019, 68(10), 1455-1465. doi: 10.1099/jmm.0.001070. [D] Fialho OB, de Souza EM, de Borba Dallagassa C, de Oliveira Pedrosa F, Klassen G, Irino K, Paludo KS, de Assis FE, Surek M, de Souza Santos Farah SM, Fadel-Picheth CM. Detection of diarrheagenic Escherichia coli using a two-system multiplex-PCR protocol. J Clin Lab Anal. 2013, 27(2), 155-61. doi: 10.1002/jcla.21578. [E]Carrillo-Casas EM, Hernández-Castro R, Suárez-Güemes F, de la Peña-Moctezuma A. Selection of the internal control gene for real-time quantitative rt-PCR assays in temperature treated Leptospira. Curr Microbiol. 2008, 56(6):539-46. doi: 10.1007/s00284-008-9096-x. |
Comments 5: In the line 182-183, the sentence “Applying the above liquid to the TSA plates and growing for 3 h for counting the number of colonies.” should be rewritten. |
Response 5: Thank you for pointing this out. I agree with this comment. I conducted the experiment again and obtained the experimental results. (Page 5, “2.7.3. Bacterial adherence and invasion assays”; Page 6, line 1-15. Page 9, Figure 6Cand Figure 6D.) |
Comments 6: In the line 205-206, “If the weight exceeds 20% of the average weight of the control group mice, the mouse would be considered dead.” Is the judgement mentioned correct? And also needs a reference. |
Response 6: Thank you for pointing this out. I agree with this comment. In previous experiments, I tested the pathogenicity of clinical isolates of Streptococcus suis types 2, 3, 7, 8, and 9 in mice. I found that mice on the brink of death were mainly characterized by coarse and messy fur, loss of appetite, weight loss, increased secretion from the corners of the eyes, hunchback, and supine position. At this point, their weight decreased by more than 20% compared to the control group mice. But the experimental datas were limited and had not been publicly published, it was incorrect to describe 'if the weight exceeds 20% of the average weight of the control group mice, the mice are considered dead'. It should be modified to “It is considered as the death of a mouse which is on the brink of death and its main symptoms include coarse and messy fur, loss of appetite, weight loss, increased secretion from the corners of the eyes, hunched back, and lying still.”. The above opinions are for your reference. (Page 6, “2.8. Animal pathogenicity tests”, line 10-13.) |
Comments 7: In the discussion, the paragraph 2 from line 353-366 should be deleted because the gene knockout method based on pSET4s is very conventional. Meanwhile, the known functions of the ATP-binding cassette transporter encoded by sugC gene in different bacteria could added in this section. |
Response 7: Thank you for pointing this out. I agree with this comment. Based on your suggestion, I have deleted this section and improved the discussion. (Page 11, “4. Discussion”, line 10-27; Page 13, “References”; [27]-[31]) |
Comments 8: Figure A1 is redundant. |
Response 8: Thank you for pointing this out. I agree with this comment. I deleted Figure A1. (Page 13,”Appendix A”) |
Reviewer 2 Report
Comments and Suggestions for Authors
This manuscript investigates the contribution of the SugC gene in Streptococcus suis to its pathogenicity. Infection experiments using SugC knockout strains have demonstrated differences in mouse survival rates, indicating that SugC contributes to pathogenicity. However, there are several concerns regarding the expressions and experimental systems described in the manuscript. Please address and revise the manuscript in response to the following comments.
The manuscript contains the expression “ABC binding transporters,” which should be corrected to “ABC transporters.” Please revise all instances of this term throughout the text. Specifically, line 66 should be corrected to “ATP-binding cassette (ABC) transporters are a type of membrane-integrated protein.”
Regarding Fig. 1, it may be omitted as it is not necessary to include the electrophoresis images of DNA fragments and plasmids used in the construction of knockout strains. A figure confirming the generation of the knockout strains should suffice.
For Fig. 4, the 16S rRNA of cΔsugC appears to be below 1.0. Please clarify this observation. Additionally, since the 16S rRNA should be used as an internal standard set to 1.0, the inclusion of the 16S rRNA graph may be unnecessary.
Fig. 5 shows bacterial count measurements; was this experiment conducted only once? Such experiments typically require at least three independent replicates. The graph should also reflect the variability of the data.
The resolution of Fig. 6 is insufficient for clearly distinguishing bacterial morphology. Either remove Fig. 6 or provide an enlarged image that clearly shows bacterial morphology.
Regarding cell adhesion and invasiveness, reference 21 does not describe the methods used. Please verify if there may be a citation error. Furthermore, the colony count was performed 3 hours after plating on TSA; is this sufficient time for colony counting? In addition, for this experiment, trypsin digestion was performed when assessing bacterial numbers for cell adhesion. This procedure could also measure bacteria that have invaded the cells, not just those adhering to the surface. The authors claim that there are differences between ΔsugC and the wild type in both cell adhesion and invasiveness experiments. However, in the experimental setup used, trypsin digestion affects both cell adhesion and invasion. Typically, trypsin digestion is not performed when evaluating cell adhesion. Please provide a reasonable justification for using trypsin digestion; otherwise, repeating the experiments may be necessary.
It is recommended to present the mouse survival experiment results as a graph rather than a table.
For lines 353-366, which discuss the generation of knockout strains using previously reported methods, detailed discussion in the Discussion section is not necessary if these methods are standard.
The discussion of SugC in Mycobacterium starting from line 367 seems to be confusingly intermingled with the experimental details of this study. Please revise the expression to clarify.
The relative expression levels of adhesion-related genes such as fbps, gdh, gapdh, and cps2J mentioned on line 393 are not shown in the results. Please either appropriately reference these results or indicate “Data not shown.”
Author Response
Comments 1: The manuscript contains the expression “ABC binding transporters,” which should be corrected to “ABC transporters.” Please revise all instances of this term throughout the text. Specifically, line 66 should be corrected to “ATP-binding cassette (ABC) transporters are a type of membrane-integrated protein.” |
Response 1: Thank you for pointing this out. I agree with this comment. Therefore, I changed “ABC binding transporters,” to “ABC transporters.” and “ATP-binding cassette transporters, also known as ATP binding transporters (ABC), were a kind of membrane integration protein commonly existing in prokaryotic and eukaryotic organisms and were one of the largest protein families in organisms “ to “ATP-binding cassette (ABC) transporters are a type of membrane-integrated protein.”.(Page 2, “1. Introduction”, line 18-20, 22-23.)
|
Comments 2: Regarding Fig. 1, it may be omitted as it is not necessary to include the electrophoresis images of DNA fragments and plasmids used in the construction of knockout strains. A figure confirming the generation of the knockout strains should suffice. |
Response 2: Thank you for pointing this out. I agree with this comment. Therefore, I have deleted the Fig. 1 and adjusted the numbering of other figures.( Page7, Results, Figures) |
Comments 3: For Fig. 4, the 16S rRNA of cΔsugC appears to be below 1.0. Please clarify this observation. Additionally, since the 16S rRNA should be used as an internal standard set to 1.0, the inclusion of the 16S rRNA graph may be unnecessary.. |
Response 3: Thank you for pointing this out. I agree with this comment. I believe that the result is caused by the error of three repeated experiments. In order to provide readers with a clearer understanding of our experimental results, I repeated the experiment and obtained the relevant results in Figure 3(This is the new number of this figure). (Page 7, “3.1. Identification of sugC gene knockout strain and complementary strain”, Figure 3.) |
Comments 4: Fig. 5 shows bacterial count measurements; was this experiment conducted only once? Such experiments typically require at least three independent replicates. The graph should also reflect the variability of the data. |
Response 4: Thank you for pointing this out. The bacterial count measurement results were obtained through three experiments conducted by the author, with three counts repeated for each group in each experiment. The average of each experimental result was taken and the Fig. 5 was plotted. The error of three repeated experiments at each time point on the graph was relatively small. By placing the graph, the error of the results could be seen. |
Comments 5: The resolution of Fig. 6 is insufficient for clearly distinguishing bacterial morphology. Either remove Fig. 6 or provide an enlarged image that clearly shows bacterial morphology. |
Response 5: Thank you for pointing this out. I agree with you. I have deleted Fig. 6. (Page 8, “3.4 Lacking the sugC gene unchange the hemolytic characteristics of TJS75 strain”.) |
Comments 6: Regarding cell adhesion and invasiveness, reference 21 does not describe the methods used. Please verify if there may be a citation error. Furthermore, the colony count was performed 3 hours after plating on TSA; is this sufficient time for colony counting? In addition, for this experiment, trypsin digestion was performed when assessing bacterial numbers for cell adhesion. This procedure could also measure bacteria that have invaded the cells, not just those adhering to the surface. The authors claim that there are differences between ΔsugC and the wild type in both cell adhesion and invasiveness experiments. However, in the experimental setup used, trypsin digestion affects both cell adhesion and invasion. Typically, trypsin digestion is not performed when evaluating cell adhesion. Please provide a reasonable justification for using trypsin digestion; otherwise, repeating the experiments may be necessary. |
Response 6: Thank you for pointing this out. I agree with you. Based on your suggestion and following the method described in reference 20, I have conducted a new experiment in this section. Indeed, as you said, trypsin can affect the adhesion effect of bacterial strains. Therefore, in future work, I will learn more from peer experts and teams, and design experiments more rigorously. (Page 5, “2.7.3. Bacterial adherence and invasion assays”; Page 6, line 1-15. Page 9, Figure 6C and Figure 6D.) |
Comments 7: It is recommended to present the mouse survival experiment results as a graph rather than a table. |
Response 7: Thank you for pointing this out. I hopes that this table could display both the death results of experimental mice and the calculated LD50 values, and using a table format may provide readers with a more intuitive presentation. If you think it is necessary to modify it into an image, I will make formatting changes to this part of the results, but may not directly display the specific value of LD50. |
Comments 8: For lines 353-366, which discuss the generation of knockout strains using previously reported methods, detailed discussion in the Discussion section is not necessary if these methods are standard. |
Response 8: Thank you for pointing this out. I agree with this comment. Based on your suggestion, I have deleted this section. (Page 11, “4. Discussion”) |
Comments 9: The discussion of SugC in Mycobacterium starting from line 367 seems to be confusingly intermingled with the experimental details of this study. Please revise the expression to clarify. |
Response 9: Thank you for pointing this out. I agree with this comment. Based on your suggestion, I have improved the discussion. (Page 11, “4. Discussion”, line 10-27; Page 13, “References”; [27]-[31] |
Comments 10: The relative expression levels of adhesion-related genes such as fbps, gdh, gapdh, and cps2J mentioned on line 393 are not shown in the results. Please either appropriately reference these results or indicate “Data not shown.” |
Response 10: Thank you for pointing this out. I agree with this comment and have added the detection results of mRNA levels of genes such as fbps, cps2J, gdh, and gapdh in the results.( Page 8-9, “3.5 Knocking out sugC gene reduces the adhesion and invasiveness of TJS75 to PK-15 cells”, Figure 6E.) |
Reviewer 3 Report
Comments and Suggestions for Authors
Vetsci-2770628
Characterization studies on the sugC gene of Streptococcus suis 2 serotype 2 in adhesion, invasion, and virulence in mice
Line 64 - … we would still lack… instead we, use “it”
Line 65 - … and its pathogenesis for
Line 105 – verify the correct order of “(Beijing Liuhe) Co., Ltd.
Line 155 - it was conducted – use passive voice
Line 205 - I didn´t understand why the 20% obese mice (average weight) were considered dead compared to the control group mice.
Lines 211 and 212 – what did you mean here:…”there was no significant between TJS75 and CΔsugC (Figure 4), but not found in ΔsugC….”
Lines 253 and 254 – instead …”was not significant difference”…. Use “…”didn´t have significant difference”...
Lines 261 and 262 - Figure 6. Gram staining and microscopic examination results. This figure does not add information, which could be very detailed in the text. Lines 277 and 278 -Figure 7A - This figure does not add information, which could be very detailed in the text.
Table 5 - something is missing?
Line 410 - It lays a foundation …

Minor errors to correct
Author Response
Comments 1: Line 64 - … we would still lack… instead we, use “it” |
Response 1: Thank you for pointing this out. I agree with this comment. Therefore, I replaced “we” with “it”.(Page 2,”1. Introduction”, line 16.) |
Comments 2: Line 65 - … and its pathogenesis for |
Response 2: Thank you for pointing this out. I agree with this comment. Therefore, I replaced “its pathogenesis of” with “its pathogenesis for”.( Page 2, ”1. Introduction”, line 17.) |
Comments 3: Line 105 – verify the correct order of “(Beijing Liuhe) Co., Ltd. |
Response 3: Thank you for pointing this out. I agree with this comment. Therefore, I has verified and corrected the information. The name of this company is BGI TECH SOLUTIONS (BEIJING LIUHE) CO., LIMITED. (Page 4, “2.2. Construction of sugC gene knockout strain and complementary strain”, line 13.) |
Comments 4: Line 155 - it was conducted – use passive voice |
Response 4: Thank you for pointing this out.I agree with this comment. Therefore, I replaced “it is conducted” with “it was conducted”.( Page 5, “2.7.1. Exploring the minimum bactericidal concentration of GEN for TJS75, ΔsugC and CΔsugC”, line 1.) |
Comments 5: Line 205 - I didn´t understand why the 20% obese mice (average weight) were considered dead compared to the control group mice. |
Response 5: Thank you for pointing this out. I agree with this comment. In previous experiments, I tested the pathogenicity of clinical isolates of Streptococcus suis types 2, 3, 7, 8, and 9 in mice. I found that mice on the brink of death were mainly characterized by coarse and messy fur, loss of appetite, weight loss, increased secretion from the corners of the eyes, hunchback, and supine position. At this point, their weight decreased by more than 20% compared to the control group mice. Because the experimental data was limited and had not been publicly published, it was incorrect to describe 'if the weight exceeds 20% of the average weight of the control group mice, the mice are considered dead'. It should be modified to “It is considered as the death of a mouse which is on the brink of death and its main symptoms include coarse and messy fur, loss of appetite, weight loss, increased secretion from the corners of the eyes, hunched back, and lying still.”. The above opinions are for your reference. (Page 6, “2.8. Animal pathogenicity tests”, line 10-13.) |
Comments 6: Lines 211 and 212 – what did you mean here:…”there was no significant between TJS75 and CΔsugC (Figure 4), but not found in ΔsugC….” |
Response 6: Thank you for pointing this out. I agree with this comment. The words were corrected to “The sugC expression was detected by the qRT-PCR with primers of qsugC-F/qsugC-R. The results showed that the expression of the sugC gene could be detected in TJS75 and CΔsugC, and there is no significant difference in expression level (Figure 3), but not found in ΔsugC.” (Page 7, “3.1. Identification of sugC gene knockout strain and complementary strain”, line 4-7.) |
Comments 7: Lines 253 and 254 – instead …”was not significant difference”…. Use “…”didn´t have significant difference”... |
Response 7: Thank you for pointing this out. I agree with this comment. Therefore, I replaced “was not significant difference” with “didn´t have significant difference”.( Page8 , line 1.) |
Comments 8: Lines 261 and 262 - Figure 6. Gram staining and microscopic examination results. This figure does not add information, which could be very detailed in the text. . |
Response 8: Thank you for pointing this out. I agree with this comment. I have deleted this figure. |
Comments 9: Lines 277 and 278 -Figure 7A - This figure does not add information, which could be very detailed in the text. |
Response 9: Thank you for pointing this out. I agree with this comment. I have deleted this figure. |
Comments 10: Table 5 - something is missing? |
Response 10: Thank you for pointing this out. I'm sorry that this data has caused you confusion. At first, I wanted to explain the meaning of "/". However, the selected annotations overlap with the data in the table. Therefore, I have corrected it to "*". (Page 10, Table 5) |
Comments 11: Line 410 - It lays a foundation … |
Response 11: Thank you for pointing this out. I agree with this comment. Therefore, I replaced “lays a foundation” with “It lays a foundation”. (Page 12, “5. Conclusions”, line 8.) |
Round 2
Reviewer 2 Report
Comments and Suggestions for Authors
The manuscript has been appropriately revised to address the reviewers' comments, and therefore, I believe it merits acceptance for publication.